# Exopolysaccharides from Marine Microbes: Source, Structure and Application

**DOI:** 10.3390/md20080512

**Published:** 2022-08-12

**Authors:** Mingxing Qi, Caijuan Zheng, Wenhui Wu, Guangli Yu, Peipei Wang

**Affiliations:** 1College of Food Science and Technology, Shanghai Ocean University, Shanghai 201306, China; 2Key Laboratory of Tropical Medicinal Resource Chemistry of Ministry of Education, College of Chemistry and Chemical Engineering, Hainan Normal University, Haikou 571158, China; 3Key Laboratory of Tropical Medicinal Plant Chemistry of Hainan Province, Haikou 571158, China; 4Key Laboratory of Marine Drugs, Ministry of Education, Shandong Provincial Key Laboratory of Glycoscience and Glycoengineering, School of Medicine and Pharmacy, Ocean University of China, Qingdao 266237, China; 5Laboratory for Marine Drugs and Bioproducts of Qingdao Pilot National Laboratory for Marine Science and Technology, Qingdao 266237, China

**Keywords:** exopolysaccharides, marine microorganisms, structure, biological activities, environmental remediation

## Abstract

The unique living environment of marine microorganisms endows them with the potential to produce novel chemical compounds with various biological activities. Among them, the exopolysaccharides produced by marine microbes are an important factor for them to survive in these extreme environments. Up to now, exopolysaccharides from marine microbes, especially from extremophiles, have attracted more and more attention due to their structural complexity, biodegradability, biological activities, and biocompatibility. With the development of culture and separation methods, an increasing number of novel exopolysaccharides are being found and investigated. Here, the source, structure and biological activities of exopolysaccharides, as well as their potential applications in environmental restoration fields of the last decade are summarized, indicating the commercial potential of these versatile EPS in different areas, such as food, cosmetic, and biomedical industries, and also in environmental remediation.

## 1. Introduction

Marine microorganisms, as important members of marine organisms, can produce numerous specific active substances in an extreme environment in low temperature, high salt, high pressure, and oligotrophic conditions. They have gained scientific interest due to their potential applications in the pharmaceutical, cosmetic and food industries, for environmental remediation, and astrobiology [1,2,3,4,5].

Exopolysaccharides (EPS), are high molecular weight carbohydrate polymers, which are secreted by microorganisms in the process of growth and metabolism [6,7,8,9,10,11,12]. The complex and diverse structure of EPS endows them with unique biological activities and functions [13,14,15,16]. Now, EPS, as a major active component of marine microbial resources [5,17], have become a new research hotspot, with advances in modern separation and analysis techniques [12,18,19,20,21,22,23,24]. First, because the EPS are produced by marine microorganisms, this is an important factor for them to survive in these extreme environments, such as high pressure, high temperatures, and high salinity [25]. Several pieces of evidence have indicated that the monosaccharide composition, high molecular weight, hydrophobicity and polycharged characteristics of marine EPS are involved in their cryoprotective effect, water-holding capacity, and good thermostability, which are essential for the survival of these psychrophiles, halophiles, and thermophiles [1,26,27,28,29,30]. In addition, the EPS from marine microbes have aroused more and more attention and are explored in the food industry, cosmetics, and biological medicine due to their unique properties, such as structural complexity, biodegradability, biological activities, and biocompatibility [31]. For example, HYD 657, also called deepsane, which was produced and secreted by the marine *Alteromonas macleodii subsp. fijiensis* biovar deepsane, has already been used in cosmetics for soothing and reducing the irritation of sensitive skin against chemical, mechanical, and UVB aggressions [32]. Levan, isolated from *H. smyrnensis* AAD6T, native and derivatives, showed high compatibility and anticoagulant activity, which activity was similar to heparin, a clinical anticoagulant [33]. A novel EPS, named as EPS-S3 from marine bacterium *Pantoea* sp., could accelerate cutaneous wound healing via the Wnt/β-catenin pathway, indicating a prospective use as a new biomaterial for skin tissue regeneration [34]. Furthermore, EPS synthesized by microorganisms have many advantages over the other polysaccharides isolated from plants, animals, and algae [22], such as high structure reproducibility, which is not possible for most plant and animal sources, due to their polysaccharide structures dependent on climate, environmental and feed conditions [21]. In addition, the high yield of EPS can be achievable by optimizing culture conditions and using the techniques of synthetic biology, which is now a promising and new research field.

At present, although the physical properties of the marine microbial EPS are promising, the studies concerning their functional and biological activities are still scarce, compared to the exploitation and application of polysaccharides from terrestrial plants and microorganisms [1]. Here, this review emphasizes the research progress in the sources, chemical structural characteristics, and application of EPS from marine microorganisms in recent years, suggesting that these marine microbial EPS can be regarded as a new and promising source, with potential applications of biological activity and environmental remediation.

## 2. The Source of Marine EPS

### 2.1. EPS Produced by General Marine Environmental Microorganisms

Among the several types of aquatic microorganisms, marine microorganisms account for half of the production of organic matter on earth [35]. Up to now, many common marine strains which can produce EPS were identified, such as *Pseudoalteromonas* species [15,36], *Bacillus* species [37], *Alteromonas* species [38], and the *Vibrio* species [39]. Several EPS with various structural features and biological activities have been isolated from those common marine strains, which are shown in Table 1 [40,41,42,43,44,45,46,47,48,49,50]. For example, an EPS, produced by a novel probiotic *Pediococcus pentosaceus* M41, was isolated from a marine source [40]. An exopolysaccharide EPS273 from marine bacterium *P. stutzeri* 273 could inhibit biofilm formation and disrupt the established biofilms of *P. aeruginosa* PAO1, indicating that EPS273 had a promising prospect in combating bacterial biofilm-associated infection [49]. A *bacterium Bacillus thuringiensis* RSK CAS4 was isolated from the ascidian *Didemnum granulatum,* in which the condition of producing EPS was optimized by the response surface method [42]. An EPS-producing strain FSW-25, assigned to the genus *Microbacterium*, was isolated from the Rasthakaadu beach, Kanyakumari, which could produce a large quantity of EPS [50]. Recently, a strain of *Bacillus cereus* was isolated from the Saudi Red Sea coast. EPSR3 was a major fraction of the EPS from this marine strain, which showed antioxidant, antitumor, and anti-inflammatory activities. These biological activities of EPSR3 may be attributed to its content of uronic acids [51].

Moreover, the halophilic bacteria are known to produce EPS for withstanding the osmotic pressure. However, due to their ubiquitous distribution in saline environments, the exploration for biologically active novel EPS from halophilic bacteria is in its early stages. A bacterial EPS, named as A101 from the strain *Vibrio* sp.QY101, had antibiofilm activity [52]. An EPS named as HMEPS was first isolated from *Halolactibacillus miurensis* and had good antioxidant activity [53]. A novel EPS, designated hsEPS, was successfully isolated from the high-salt fermented broth of a novel species, *Halomonas saliphila* LCB169T. The structural of hsEPS was well-characterized as having a major backbone composed of (1→2)-linked α-D-Man*p* and (1→6)-linked α-D-Man*p*, with branches substituted at C-2 by T-α-D-Man*p* and at C-6 by the fragment of T-α-D-Man*p*-(1→2)-α-D-Man*p*-(1→ [54].

### 2.2. EPS Produced by Polar Microorganisms

Most microorganisms in the deep-sea and polar environment are affected by low temperatures and poor nutrition [64]. For the microorganisms living in a cold environment, it has been shown that their cells are almost surrounded by EPS [5]. The ability of organisms to survive and grow in a low-temperature environment depends on a series of adaptive strategies, including membrane structure modification. To understand the role of the membrane in adaptation, it is necessary to determine the cell wall components that represent the main components of the outer membrane, such as EPS. Studies have indicated that the secreted EPS with negatively charged residues, such as sulfate and carboxylic groups, allowed them to form hydrated viscous three-dimensional networks that confer adhesive and barrier properties against freezing temperatures [1]. For example, Ornella Carrión et al. isolated *Pseudomonas* ID1 from the marine sediment samples of Antarctica. The EPS produced by this strain showed significant protection against the cold [65]. The *Pseudomonas* sp. BGI-2 isolated from the glacier ice sample could produce high amounts of EPS, which had cryoprotective activity [66]. The produce condition of EPS from a cold-adapted *marinobacter*, namely as W1-16, was optimized by evaluating the influences of the carbon source, temperature, pH and salinity. The monosaccharide composition of this EPS resulted in Glc:Man:Gal:GalN:GalA:GlcA, with a relative molar ratio of 1:0.9:0.2:0.1:0.1:0.01 [67]. Now, several EPS isolated from psychrophilic bacteria in the Arctic and Antarctic marine environment have been reported, which have a potential application in the cryopreservation, food, and biomedical industries [21,26,55,56,57,58,59,60]. The source, structural, and biological information of these EPS are shown in Table 1.

### 2.3. EPS from Marine Hot Spring Microorganisms

Over the past decade, lots of microbes from marine hot springs have been reported, most of which produce special EPS to protect themselves from extreme conditions. These thermophilic microorganisms are classified as thermophiles growing at 55 °C~80 °C and hyperthermophiles growing above 80 °C. The thermophilic microorganisms contain multiple genera, such as *Aeribacillus, Anoxybacillus, Brevibacillus, and Geobacillus* [31]. The EPS produced by thermophilic bacteria usually have a high molecular weight with good emulsifying properties, leading to great potential application in the food and cosmetics industries [19,28,61,62,63]. Four thermophilic aerobic Bacillus isolated from Bulgarian hot springs are reported by Radchenkova et al., which are *Aeribacillus pallidus*, *Geobacillus toebii*, *Brevibacillus thermoruber*, and *Anoxybacillus kestanbolensis*. These bacteria can all produce EPS. After optimizing the culture conditions of the *Aeribacillus pallidus* strain 418, the output of EPS1 and EPS2 has more than doubled [31]. A novel exopolysaccharide RH-7 with a high molecular weight of 2000 kDa was produced by this marine bacterial strain assigned to the genus R*hodobacter* from the surface of the marine macroalgae (*Padina* sp.). This EPS showed high-temperature resistance and could act as a bio-emulsifier to create a high pH and temperature-stable emulsion of hydrocarbon/water [61].

## 3. The Structural Characteristics of Marine EPS

### 3.1. Structural Characterization Methods of Marine Microbial EPS

The marine microbial EPS have high structural diversity and complexity. To evaluate their structure, the monosaccharide composition, molecular weight, and glycosidic linkage need to be determined. Before the structural analysis, a homogeneous EPS should be obtained to remove the influence of other salt, pigment, and protein impurities. At present, the commonly used purification methods of EPS include ethanol precipitation, ultrafiltration, ion-exchange chromatography, and gel chromatography. The methods of SDS-PAGE and DOC-PAGE with Alcian blue staining are very useful to detect the presence of EPS [68]. The monosaccharide composition of EPS has been determined by a variety of methods, including acid hydrolysis followed by appropriate derivatization and gas chromatography (GC); pre-column derivatization with high-performance liquid chromatography (HPLC); and high-performance anion-exchange chromatography with pulsed amperometric detection (HPAEC-PAD) [69,70]. The molecular weight can be determined by high-performance gel-permeation chromatography (HPGPC), combined with differential detector (ID) or multi-angle laser light scattering (MALS) [71]. Furthermore, through the data of the Fourier-Transform infrared spectroscopy (FTIR), methylation analysis, and nuclear magnetic resonance (NMR), the glycosidic bond-linking types and main functional groups can be obtained [26,72,73,74].

Besides these classical chemical procedures, new and powerful tools, such as zeta potential and particle-size analyzer, attenuated total reflectance Fourier-Transform infra-red spectroscopy (ATR-FTIR), differential scanning calorimetry (DSC), scanning electron microscope (SEM), atomic force microscopy (AFM), circular dichroism spectrum (CD), small-angle neutron scattering (SANS), and X-ray diffraction (XRD) techniques have been applied to investigate the surface morphology and physical properties of EPS [28,40,41,75,76,77,78]. For example, the physicochemical properties and rheological properties of an EPS-M41, produced by a novel probiotic *Pediococcus pentosaceus* M41 isolated from a marine source, were evaluated in detail. The average molecular weight of this EPS was determined to be 682.07 kDa by HPGPC method. The EPS-M41 consisted of Ara, Man, Glc, and Gal with a molar ratio of 1.2:1.8:15.1:1.0 by the GC method. The structure of the EPS-M41 was proposed as →3) α-D-Glc (1→2) β-D-Man (1→2) α-D-Glc (1→6) α-D-Glc (1→4) α-D-Glc (1→4) α-D-Gal (1→), with Ara linked at the terminals by FTIR and NMR analysis. The SEM analysis showed that the EPS-M41 possessed a unique compact, stiff and layer-like structure. The particle and zeta charges analyses exhibited that the EPS-M41 had a size diameter of 446.8 nm and a zeta potential of -176.54 mV. The DSC thermogram exhibited that the EPS-M41 had a higher melting point, indicating its resistance to the thermal processes [40]. Another example, the ATR-FTIR technique, was used to observe the movement of the -SH, -PO4, and -NH functional groups in the EPS from *Pseudomonas pseudoalcaligenes* NP103, and confirmed their involvement in the Pb (II) binding. The results emphasized the potential importance of *P. pseudoalcaligenes* NP103 EPS as a biosorbent for the removal of Pb (II) from the contaminated sites [28].

### 3.2. Examples of Marine Microbial EPS in the Last Decade

Several reviews have summarized the culture and fermentation conditions, distribution, biosynthesis, and biotechnological production of microbial EPS from marine sources [1,12,14,19,20,21,79,80,81]. In the past decade, with the development of separation and identification technology, numbers of new marine microorganisms have been identified. By optimizing the culture conditions, novel EPS with new biological activities have been discovered. Here, a variety of EPS obtained from marine microorganisms, including bacteria, fungi, and microalgae, in the last decade are summarized in Table 1, Table 2 and Table 3. It will give us more useful information of the structure–activity relationship of the marine EPS through the analysis of their origin, monosaccharide composition, molecular weight, and bioactivities.

The examples of EPS obtained from marine bacteria in the last decade are shown in Table 1. The marine microbial EPS are complex, and large polymers, usually composed of more than one monosaccharide, including pentoses, hexoses, amino sugars, and uronic acids. More commonly, they attach to proteins, lipids, or non-carbohydrate metabolites, such as pyruvate, sulphate, acetate, phosphates, and succinate, as their additional structural components, which increased their structural diversity and complexity [13]. Some of the monosaccharides, such as fucose, ribose, uronic acid and aminosaccharides, which are not common in plant polysaccharides, are widely found in the marine bacterial EPS. These special monosaccharides present are also closely related to the biological functions of these marine bacterial EPS. For instance, the structure of a novel EPS isolated from *Colwellia psychrerythraea* 34H was fullly characterized to have a repeating unit, composed of a N-acetyl-quinovosamine (QuiN) unit and two galacturonic acid residues both decorated with alanine amino acids, which had a significant cryoprotective effect. By NMR and computational analysis, the pseudo-helicoidal structure of this EPS may block the local tetrahedral order of the water molecules in the first hydration shell, and could inhibit the ice recrystallization [26]. A novel anionic EPS, named as FSW-25, was produced by marine *Microbacterium aurantiacum.* FSW-25 was a high molecular-weight heteropolysaccharide with a high uronic acid content. It had good antioxidant potential when compared with xanthan, which might be due to the presence of sulphate and its higher uronic content [50]. In addition, the monosaccharide composition of the EPS and other residues could play an essential role in thermostability. For example, the high thermal stability of EPS1-T14 produced by *Bacillus licheniformis* was mainly attributed to the fucose content [28,29]. The role of the monosaccharides’ composition in the thermal stability of the marine bacteria EPS has to be further investigated.

Compared with the diversity and complexity of the marine bacteria EPS, these EPS isolated from marine fungi exhibit less significant diversity in the monosaccharide composition (Table 2). The monosaccharide compositions show that these marine fungal EPS are mainly composed of neutral monosaccharide, including Glc, Man, and Gal with a different molar ratio. Generally, these EPS have antioxidant activity. Mao et al. completed relatively systematic studies on the structure and activity screening of marine fungal EPS. Several of the EPS are isolated and fullly characterized from *Aspergillus versicolor*, *Aspergillus Terreus, Fusarium oxysporum,* and *Hansfordia sinuosae* (Table 2). Recently, a novel EPS (AUM-1) with immunomodulatory activity was obtained from the marine *Aureobasidium melanogenum* SCAU-266. The AUM-1 with a molecular weight of 8000 Da had a main monosaccharide of Glc (97.30%), whose structure possessed a potential backbone of α-D-Glcp-(1→2)-α-D-Man*p*-(1→4)-α-D-Glc*p*-(1→6)-(-(SO_3_^−^)-4-α-D-Glc*p*-(1→6)-1-β-D-Glc*p*-1→2)-α-D-Glc*p*-(1→6)-β-D-Glc*p*-1→6)-α-D-Glc*p*-1→4)-α-D-Glc*p*-6→1)-[α-D-Glc*p*-4]_26_→1)-α-D-Glc*p* [74]. The possible structure of these marine fungal EPS mentioned in Table 2 are shown in Figure 1.

Besides the marine bacteria and fungi, marine microalgae and cyanobacteria are other important resources to produce EPS. Information of some of the EPS obtained from marine microalgae and cyanobacteria in the last decade are shown in Table 3. These EPS usually have a complex monosaccharide composition with uronic acid and sulfate groups, with various biological activities such as antioxidant, antiviral, antifungal, antibacterial, anti-ageing, anticancer, and immunomodulatory activities [89,90]. Recently, Esqueda et al. systematically explored the diversity of 11 microalgae strains belonging to the proteorhodophytina subphylum for EPS production. Regarding the compositions, some of the common features were highlighted, such as the presence of Xyl, Gal, Glc, and GlcA in all of the compositions, but with different amounts depending on the samples. In addition, the existence of sulfate groups in EPS from those microalgae strains were much more different [91]. The EPS from *Chlorella sorokiniana* had anticoagulant and antioxidant activities. The sulfate content and their binding site, monosaccharide composition, and glycoside bond were involved in its bioactivity [92]. Cyanoflan, a cyanobacterial-sulfated EPS, was characterized in terms of its morphology, structural composition, and rheological and emulsifying properties. The glycosidic linkage analysis revealed that this EPS had a highly branched complex structure with a large number of sugar residues, including Man, Glc, uronic acids, Gal, Rha, Xyl, Fuc, and Ara with a molar ratio of 20:20:18:10:9:9:8:6. The high molecular weight (>1 MDa) and entangled structure was consistent with its high apparent viscosity in aqueous solutions and high emulsifying activity [75]. The EPS from the cyanobacterium *Nostoc carneum* was a type of polyanionic polysaccharide that contained uronic acid and sulfate groups [93]. Another EPS from *Tetraselmis suecica* (Kylin) with antioxidant and anticancer activities also had a high amount of uronic acid [94]. The acid groups played important roles in the antioxidant activity of the marine EPS.

## 4. The Biological Activities of Marine EPS

The structure of the marine microbial EPS is complex and diverse [101], and is linked to various biological activities, such as antibacterial, antioxidant, anti-cancer, antifreeze, anti-inflammatory, enhancement of immune activity and blood pressure, and lipid reduction [102,103,104,105]. In addition, due to the particularity of the marine microbial environment, the EPS produced by marine microbials also have a potential application value in the marine ecological environment. Here, we mainly introduced the antioxidant activity, anticancer activity, anti-infectious activity, and immune-enhancing biological activity of the marine microbial EPS (as shown in Table 1, Table 2 and Table 3 and Figure 2) and their potential application in bioremediation and carbon sequestration.

### 4.1. Antioxidant Activity

Oxygen is the key substance in the normal life-metabolism of aerobic organisms [106]. In the metabolism of organisms, the living organisms inevitably produce reactive oxygen species (ROS) [107]. High levels of ROS may disrupt the pro-oxidant/antioxidant balance in organisms, leading to oxidative stress [108]; excessive ROS will destroy the normal function of lipids, proteins, and DNA in human cells, thus inducing various diseases [109]. Several pieces of evidence have proved that antioxidants play an important role in protecting humans from cancer, diabetes, cardiovascular disease, and neurodegenerative diseases related to different types of oxidative damage [110,111,112].

The EPS from marine bacteria usually have strong antioxidant activity, which is related to the structural characteristics of EPS, including sulfate content and their binding site, monosaccharide residues, and glycoside bonds [13]. The marine bacteria EPS usually contribute to the formation of biofilms, thus adapting to extreme environments, such as high salinity, low temperatures, and high osmotic pressure [113,114,115]. For example, the AEPS from *Rhodella reticulata* have a stronger scavenging ability of superoxide anions than that of the standard antioxidant, α -tocopherol [116]. The EPS isolated from the Arctic marine Bacterium *Polaribacter* sp. SM1127 have a good antioxidant capacity. The antioxidant capacity is significantly higher than that of hyaluronic acid (HA), a common free-radical scavenging adhesive in cosmetics, which indicated that this EPS has good application prospects in the future cosmetics’ antioxidant field [59]. Further, SM1127 could remove the excess ROS produced by wound infection and inflammation, thus accelerating wound healing. Therefore, this EPS is likely to be used to accelerate the healing of frostbite, burns, and other wounds [60]. Wu et al. reported a marine bacterial EPS produced by *P. stutzeri 273* named EPS27 with a good clearance rate of hydroxyl radicals, which could reach 70% when the concentration of EPS is 60 μg/mL. Therefore, EPS27 has good antioxidant activity and has potential application prospects in the food and health-care fields [49]. The skin is the largest organ of the human body and skin wound healing is an important clinical problem [117,118,119]. Since synthetic drugs have a high risk of adverse effects, such as allergies and drug resistance, natural products such as EPS are becoming increasingly important and are strongly recommended as alternative medicines for wound healing.

Another source of antioxidants is EPS produced by marine fungi. Wang et al. found a new extracellular polysaccharide (YSS) from the marine fungus *Aspergillus Terreus*, which was composed of Man and Gal units with a molecular weight of 18.6 kDa [85]. YSS had a strong scavenging ability on DPPH radicals, and the EC50 was 2.8 mg/mL. Chen et al. reported that the marine fungus, *Fusarium oxysporum*, produced a novel galactofuranose-containing EPS Fw-1, which was mainly composed of Gal, Glc, and Man with a molecular weight of 61.2 kDa [83]. The EC50 of Fw-1 on the scavenging of hydroxyl and superoxide radicals was 1.1 and 2.0 mg/mL, respectively, which was larger than that of the EPS named as AVP isolated from the marine *Aspergillus Versicolor* LCJ-5-4 (EC_50_ is 4.0 mg/mL). The antioxidant EPS extracted from marine fungi had a relatively simple monosaccharide composition and a small molecular weight, which was more suitable for studying the relationship between the structure of the marine polysaccharides and antioxidants [80,120].

The epidemiological investigations have proved a strong correlation between the antioxidant utilization and a diminished risk of commonplace chronic diseases, such as cardiovascular disease and cancer. Compared with many reports on the antioxidant activity of marine microbial EPS, there are few reports on the separation, purification, and structure analysis of marine microbial EPS. It is necessary for the in-depth study of their structure–activity relationship.

### 4.2. Anti-Cancer Activity

Now, new sources of non-toxic natural substances with potential anticancer effects, are being actively investigated [121]. In the past decade, there has been a great deal of interest in the development of anti-cancer polysaccharide drugs. The marine microorganisms have unique metabolic and physiological abilities, which give them the ability to produce various biological compounds [122,123,124], such as EPS. Several marine microbial EPS have been reported to have anticancer activity by the dysfunction of mitochondria, the inhibition of cell proliferation, or the modulation of the immune system [42,58,94,125,126,127,128]. Matsuda et al. studied the marine *Pseudomonas* polysaccharide B1, and found that it could induce U937 cells’ apoptosis [128]. Chen et al. reported that the Antarctic bacterium *Pseudoaltermonas* sp. S-5 produced a hetero-exopolysaccharide (named PEP) which could significantly inhibit human leukemia cell K562 growth [58]. Moreover, Ramamoorthy Sathishkumar et al. found that EPS from ascidian symbiotic bacterium *Bacillus thuringiensis* had good anticancer activity in vitro. This polysaccharide showed potential cytotoxicity against the cancer cell lines A549 and HEP-2 compared with normal Vero cells. The inhibition rate of EPS on both of the cancer cell lines increased in a dose-dependent manner [42]. The AS2-1 produced by *Alternaria* sp. could also inhibit the growth of Hela, HL-60, and K562 cells in a concentration-dependent manner [84]. The marine bacterial exopolysaccharide EPS11 could effectively inhibit the adhesion, migration, and invasion of hepatocellular carcinoma cells; this underlying target protein and molecular mechanism was first explored through the β 1-integrin signal pathway by targeting type I collagen [126]. Recently, one study showed that the newly isolated marine bacterial EPS could enhance antitumor activity in HepG2 cells by affecting the key apoptotic factors and activating the toll-like receptors (TLR) [129]. Other studies have shown that the chemical modification of EPS, such as acetylation, carboxymethylation, and sulfonation, can also enhance its biological activity [130,131], and then enhance its anticancer activity. Mazza et al. confirmed that the two polysaccharides, EPS-DR and EPS-DRS, could form complexes with scandium, and that these complexes showed a variety of biological activities, especially antiproliferative properties in cancer cells.

### 4.3. Anti-Infectious Diseases

EPS also play an important role in fighting infectious diseases. A quantity of research evidenced that the immune activity and anti-viral activity of marine microbial EPS have potential value in inhibiting some influenza viruses and bacteria [132]. EPS, as a potent antibacterial agent, can inhibit bacterial growth mainly by inhibiting biofilm formation. Mikhlid H et al. reported that *Enterobacter* sp. ACD2 EPS from the Tabuk region of Saudi Arabia had a certain inhibitory effect on *E. coli* and *Staphylococcus aureus* [48]. Durairajan Rubini et al. reported a marine polysaccharide with good antibacterial activity and strong inhibition against uropathogenic *Escherichia coli* (UPEC), providing an antibiotic-free method for the treatment of urinary tract infections [133]. Similarly, Wu et al. reported that an exopolysaccharide EPS273 from the culture supernatant of the marine bacterium *P. stutzeri* 273 inhibited *Pseudomonas aeruginosa* by anti-biofilm activity [49]. It can be effective not only in animal bacteria, but also in plant bacteria. Marwa Drira et al. found that EPS produced by *Porphyridium sordidum* had led to the control of fungal growth by the plant, and EPS could act as an inducer to enhance the resistance of A. thaliana to *F. oxysporum* [24]. In addition, the sulfated EPS from *Porphyridium* sp. has shown an antiviral effect against the herpes viruses (HSV-1 and HSV-2) [134,135].

### 4.4. Immunomodulatory Activity

The main function of the immune system is to identify and eliminate pathogens to maintain physiological balance and stability [136]. When the immunity is impaired, it will lead to various adverse immune reactions [137]. Several of the EPS synthesized by marine microorganisms have immunomodulatory activities [45,60,61,74,138,139,140]. For example, an EPS named as EPS2E1 was extracted from marine *Halomonas* sp. and displayed good immune-enhancing activity, mainly by activating the MAPK and NF-κB pathways [56]. Soumya Chatterjee et al. reported that sphingobactan, a new α-mannan EPS from Arctic *Sphingobacterium* sp. IITKGP-BTPF3 significantly reduced the NO production of LPS-induced macrophages. These results indicated that sphingobactan had a potential activation effect on the anti-inflammatory effects of macrophages in vitro [57]. On the one hand, evidence has shown that the expression of cytokines, such as interleukin (IL), tumor necrosis factor (IF-α), and interferon, can be induced by marine microbial EPS [141]. Moreover, Adriana et al. reported that EPS-1 could contribute to improve the immune surveillance of PBMC toward viral infection, by triggering a polarization in favor of the Th1 subset. EPS-1 produced by *Bacillus Licheniformis* could induce the production of cytokines to enhance immune regulation [138]. It mainly promoted the macrophages to secrete the mediators and enzymes that play a vital role in mediating inflammation and tissue repair, such as NO, COX-2, IL-1, IL-6, and TNF-α in the RAW264.7 macrophages [142,143,144,145]. YCP, a natural EPS from the mycelium of the marine filamentous fungus *Phoma herbarum* YS4108, could bind to TLR-2 and TLR-4 and had great antitumor potential via exhibiting a specific immunomodulatory capacity, mediated by T cells and dendritic cells (DCs) [146]. Recently, a novel EPS (AUM-1) was obtained from marine *Aureobasidium melanogenum* SCAU-266, with a potential effect on the ferroptosis-related immunomodulatory property in RAW264.7 cells. The mechanism studies have shown that it can adjust the expression of GPX4, regulate glutathione (oxidative), and directly cause lipid peroxidation, due to the higher ROS level by the glutamate metabolism and TCA cycle (Figure 3) [74].

## 5. Environmental Remediation

### 5.1. Biological Restoration

With the progress of industrialization, wastewater treatment is particularly important. Traditional wastewater treatment methods include mixing, precipitation, filtration, or chlorine disinfection, but these methods are expensive and prone to secondary pollution [139,140,147]. The biological treatments, by contrast, are environmentally friendly. One of the important biological systems’ methods is the use of marine microbial EPS flocculation [22]. These EPS have a significant impact on the physical and chemical properties of microbial aggregates [18], which is conducive to flocculation and the precipitation of impurities in sewage. It has also been reported that the EPS interact with cells to form a large network that protects them from dehydration and toxins [148]. In addition, the marine microbial EPS are often polyanionic with uronic acids, sulfated units, and phosphate groups. These chemical groups provide EPS with a negative charge allowing them to act as ligands toward dissolved cations, as well as trace and toxic metals [1]. For example, Zhou et al. reported that Bsi 20310, produced by *Pseudoalteromonas* SP in the ocean, could effectively enhance the rate of flocculation formation in the ferric chloride coagulation RX-3B simulation of fuel wastewater [149]. Similarly, Yang et al. also reported a positive correlation between LB-EPS and the sludge flocculation rate [150]. Likewise, microbial EPS are also valuable in dealing with other environmental problems. Flocculation is a common method used to remove heavy metals and other pollutants from sewage. Likewise, Paritosh Parmar et al. reported on the EPS produced by PBR1 and PBL1, collected from a fish, Indian mackerel, and an Indian squid. These two kinds of bacteria, PBR1 and PBL1, are luminescent bacteria, subordinate to vibrio genera. Although most of the vibrio strains are pathogenic, the results of a β hemolysin test for PBR1 and PBL1 showed that these two strains are not pathogenic, and that the EPS produced by them can be used to repair heavy metal pollution in the water environment, and that the luminescence characteristics of these two strains can be used to develop luminescent biosensors [151]. Given the important role of EPS in wastewater treatment, Salama et al. proposed five better ways to understand EPS: (1) Development of EPS extraction methods; (2) Establishment of EPS in situ analytical methods; (3) Identifying the sub-fractions of EPS; (4) Elucidation of the key roles of EPS; (5) Identification of the key factors influencing the production of EPS [18].

### 5.2. Oil pollution Remediation

There have been related applications of marine EPS in dealing with the leakage of offshore crude oil [152]. Noriyuki et al. found that the EPS produced by *Monascus* could be used to deal with the offshore oil spill and accelerate the degradation of crude oil [153]. For example, Ant-3b could emulsify n-hexadecane at low temperatures and had great potential in solving crude oil pollution in a low-temperature environment [154]. At the same time, the research of Ilori, Amobi, and Odocha showed that biosurfactants are superior to synthetic surfactants [155].

### 5.3. Carbon Sequestration

Over the past decade, the oceans have absorbed a quarter of the Earth’s greenhouse gas emissions through the carbon (C) cycle, a naturally occurring process. All of the aspects of the ocean C cycle are now being incorporated into climate change mitigation and adaptation plans [156]. According to the Global Biodiversity Assessment by the United Nations Environment Program, there are 178,000 marine species which belong to 34 phyla [157]. Thus, the ocean’s biodiversity represents 50% of the whole globe’s biodiversity [158,159]. Marine primary productivity is handled by heterotrophic microorganisms; the substrate specificity of extracellular enzymes, the rate at which they function in seawater and sediments, and the factors that control their production, distribution, and active life span, are central to the carbon cycle in marine systems [160].

The EPS produced by marine microorganisms are significant for the carbon cycle and have also been reported in the deep ocean. Joseph et al. reported that the EPS produced by the marine psychrophilic bacterium *Colwellia psychrerythraea* strain 34H could affect the carbon cycle and nutrient transfer in the deep sea [161]. In the Antarctic environment, the EPS promote the accumulation of soil organic matter at the landscape level in depauperate Antarctic soil [79,162]. Antarctic green algae sequester 479,000 tons of carbon each season [163]. The EPS of microorganisms in the Antarctic environment, which exist in the form of cell aggregates, biofilms, microbial mats, biocrusts, and so on, influence the global carbon cycle [164,165].

## 6. Conclusions and Future Perspectives

The investigation of marine environments, their microbial diversity, and unique polysaccharides are now becoming one of the focuses of scientific research. In the past decade, scientific research has focused on the isolation and culture of marine microorganisms and their unique biological activities. The EPS from marine microorganisms have considerable potential in bioremediation, wastewater treatment, heavy metal treatment, and marine oil pollution. Due to its unique and complex macromolecular structure, the marine microbial EPS show great application prospects in the pharmaceutical and biomedical fields, such as antifreeze, anti-oxidation, anti-cancer, anti-inflammation, antibacterial, and so on. Although the EPS from marine microorganisms have good applications in many fields, there are relatively few studies on the marine microbial EPS, especially in the study of EPS produced by the extreme marine environment, partly due to the difficulty of isolation and culture conditions. In brief, new and updated technical strategies are needed to constantly achieve the separation, purification, and structural analysis of novel marine microbial EPS. Furthermore, the biological activities and structure–activity relationship of these EPS need further exploration.

## Figures and Tables

**Figure 1 marinedrugs-20-00512-f001:**
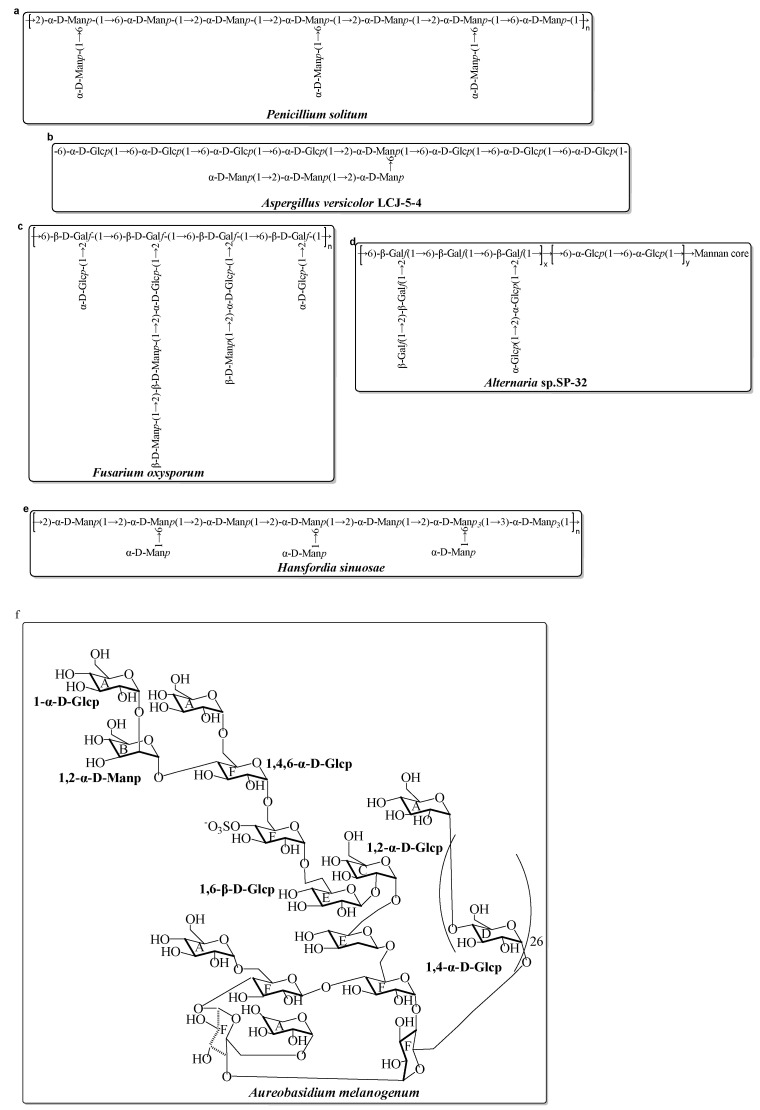
Structures of several EPS, isolated from marine fungi in the last decade [74,83,84,85,86,87,88]. (**a**) EPS from *Penicillium solitum*; (**b**) EPS from *Aspergillus versicolor*; (**c**) EPS from *Fusarium oxysporum*; (**d**) EPS from *Alternaria* sp.; (**e**) EPS from *Hansfordia sinuosae*; (**f**) EPS from *Aureobasidium melanogenum*.

**Figure 2 marinedrugs-20-00512-f002:**
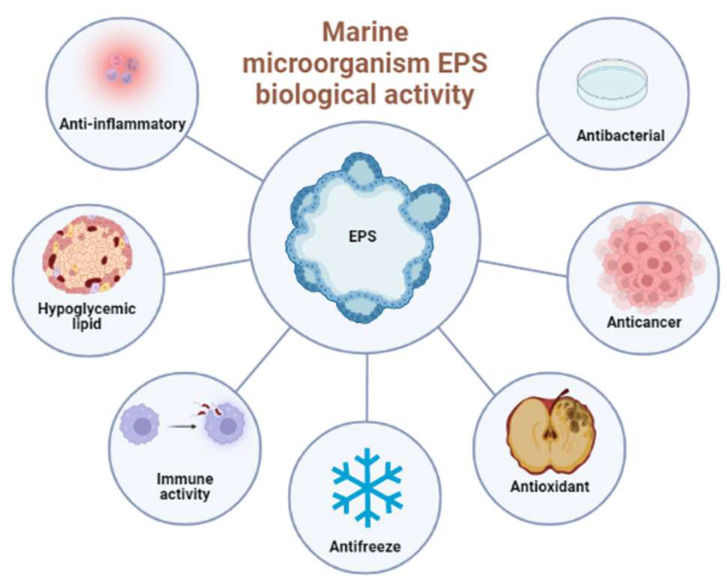
Main biological activities of EPS from marine microorganisms.

**Figure 3 marinedrugs-20-00512-f003:**
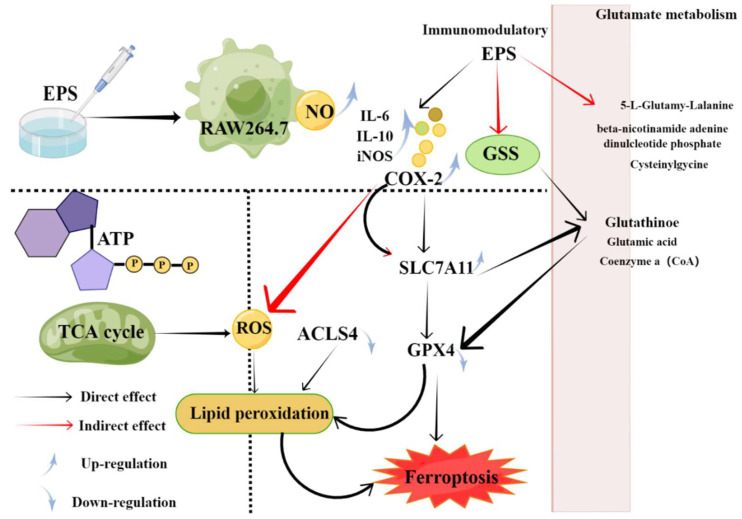
Potential mechanisms of ferroptosis-related immunomodulatory of AUM-1 [74].

**Table 1 marinedrugs-20-00512-t001:** Information of some EPS obtained from marine bacteria in the last decade.

Source	Name	Preparation Method	Monosaccharides Composition	Mw (Da)	Bioactivity	References
*Pantoea* sp. *YU16-S3*	EPS-S3	Ethanol precipitation extraction and purification by Sephacryl S500-HR column	Glc, Gal, GalNAc, GalN (1.9:1:0.4:0.02)	1.75 × 10^5^	Promotion of Wound Healing	[34]
*Pediococcus pentosaceus M41*	EPS-M41	Culture centrifugal extraction and purification by ultra-filtration	Ara, Man, Glc, Gal (1.2:1.8:15.1:1.0)	6.8 × 10^5^	Antioxidant, Anticancer	[40]
*Bacillus cereus KMS3-1*	EPS	Culture centrifugal extraction and purification by dialysis	Man, Glc, Xyl, Rha (73.51:17.87:2.18:6.49)		Waste-water treatment	[41]
*Oceanobacillus iheyensis*	EPS	Ethanol precipitation and purification by dialysis	Man, Glc, Ara (47.78:29.71:22.46)	2.14 × 10^6^	Anti-biofilm	[43]
*Bacillus thuringiensis* RSK CAS4	EPS	Culture centrifugal extraction and purification by Sepharose 4-LB Fast Flow column	Fuc, Gal, Xyl, Glc, Rha, Man (43.8:20:17.8:7.2:7.1:4.1)		Antioxidant, Anticancer	[42]
*Pseudoalteromonas, MD12-642*	EPS	Culture centrifugal extraction and purification by ultra-filtration	GalA, GlcA, Rha, GlcN (41–42:25–26:16–22:12–16)	>1.0 × 10^6^		[44]
*Bacillus* sp. H5	EPS5SH	Aqueous extraction and purification by GPC	Man, GlcN, Glc, Gal (1.00:0.02:0.07:0.02)	8.9 × 10^4^	Immunomodulatory activity	[45]
*Alteromonas* sp. *JL2810*	EPS	Ethanol precipitation extraction and purification by DEAE column	GalA, Man, Rha (1:1:1)	>1.67 × 10^5^		[46]
*Pseudoalteromonas* sp. YU16-DR3A	EPS-DR3A	Culture centrifugal extraction and purification by dialysis	Fuc, Erythrotetrose, Glc, Rib (6.7:1.0:1.5:1.0)	2 × 10^4^	Antioxidant	[47]
*Enterobacter* sp. *ACD2*	EPS	Culture centrifugal extraction	Glc, Gal, Fuc, GlcA (25:25:40:10)		Antibacterial	[48]
*P. stutzeri* 273	EPS273	Culture centrifugal extraction and purification by GPC	GlcN, Rha, Glc (35.4:28.6:27.2)	1.9 × 10^5^	Antibiofilm, Anti-Infection	[49]
*Microbacterim* FSW-25	EPS Mi25	Culture centrifugal extraction and purification by dialysis	Glc, Man, Fuc, GlcA	7.0 × 10^6^	Antioxidant	[50]
*Bacillus cereus*	EPSR3	Culture centrifugal extraction	Glc, GalA, Arb (2.0: 0.8: 1.0)		Antioxidant, Antitumor, Anti-inflammatory activities	[51]
*Vibrio* sp. QY101	A101	Ethanol precipitation extraction and purification by GPC	GlcA, GalA, Rha, GlcN (21.47:23.05:23.90:12.15)	5.46 × 10^3^	Antibacterial	[52]
*Halolactibacillus miurensis*	EPS	Culture centrifugal extraction and purification by Sepharose 4-LB Fast Flow column	Gal, Glc (61.87:25.17)		Antioxidant	[53]
*Halomonas saliphila* LCB169T	hsEPS	Ethanol precipitation, anion-exchange and gel-filtration chromatography	Man, Glc, Ara, Xyl, Gal, Fuc (81.22:15.83:1.47:0.59:0.55:0.35)	5.133 × 10^4^	Emulsifying activity	[54]
*C.psychrerythraea 34H*	EPS	Culture centrifugal extraction and purification by QFF column	QuiN, GalA (1:2)		Antifreeze	[26]
*Issachenkonii*	SM20310	Ethanol precipitation extraction and purification by DEAE column	Rha, Xyl, Man, Gal, Glc, GalNAc, GlcNAc (2.1:0.9:71.7:9.0:10.7:1.5:4.0)	>2.0 × 10^6^	Anti-freeze	[55]
*Halomonas* sp. 2E1	EPS2E1	Culture centrifugal extraction and purification by DEAE column and Sephadex G75 column	Man, Glc (3.76:1)	4.7 × 10^4^	Immunomodulatory activity	[56]
*Sphingobacterium* sp. *IITKGP-BTPF3*	Sphingobatan	Culture centrifugal extraction and purification by DEAE column	Man	>2 × 10^6^	Immunomodulatory activity	[57]
*Pseudoaltermonas* sp.	PEP	Culture centrifugal extraction and purification by dialysis and GPC	Glc, Gal, Man (4.8:50.9:44.3)	3.97 × 10^5^	Anticancer	[58]
*Polaribacter* sp.	SM1127 EPS	Ethanol precipitation extraction and purification by Sepharose column	Rha, Fuc, GlcA, Man, Gal, Glc, GlcNAc (0.8:7.4:21.4:23.4:17.3:1.6:28.0)	2.2 × 10^5^	Promotion of Wound Healing, Prevention of Frostbite Injury, Antioxidant	[59,60]
*Aeribacillus pallidus 418*	EPS1, EPS2	Culture centrifugal extraction and purification by Sepharose DEAE CL-6B column	Man, Glc, GalN, GlcN, Gal, Rib (69.3:11.2:6.3:5.4:4.7:2.9); Man, Gal, Glc, GalN, GlcN, Rib, Ara (33.9:17.9:15.5:11.7:8.1:5.3:4.9)	7 × 10^5^; >1 × 10^6^		[31]
*Rhodobacter johrii CDR-SL 7Cii*	EPS RH-7	Ethanol precipitation and purification by dialysis	Glc, GlcA, Rha, Gal (3:1.5:0.25:0.25)	2 × 10^6^	Emulsifying activity	[61]
*Alteromonas ininus*	GY785	Culture centrifugal extraction and purification by ultra-filtration	Rha, Fuc, Man, Gal, Glc, GalA, GlcA (0.2:0.1:0.4: 3.6:4.7:1.0:2.0)	2.0 × 10^6^		[62,63]

**Table 2 marinedrugs-20-00512-t002:** Information of some EPS obtained from marine fungi in the last decade.

Source	Name	Preparation Method	Monosaccharides Composition	Mw (Da)	Bioactivity	References
*Aureobasidium* *melanogenum* SCAU-266	AUM-1	Alcohol precipitation and further purified through DEAE-column	Glc, Man, Gal (97.30:1.9:0.08)	6.0 × 10^3^	Immunomodulatory activity	[74]
*Aspergillus Terreus*	YSS	Culture centrifugal extraction and purification by QFF column	Glc, Man (8.6:1.0)	1.86 × 10^4^	Antioxidant	[82]
*Fusarium oxysporum*	Fw-1	Culture centrifugal extraction and purification by QFF column	Gal, Glc, Man (1.33:1.33:1.00)	6.12 × 10^4^	Antioxidant	[83]
*Alternaria* sp.	AS2-1	Culture centrifugal extraction and purification by QFF column	Man, Glc, Gal (1.00:0.67:0.35)	2.74 × 10^4^	Anticancer, Antioxidant	[84]
*Aspergillus versicolor*	AWP	Culture centrifugal extractionand purification by QFF column	Glc, Man (8.6:1.0)	5 × 10^7^		[85]
*Aspergillus versicolor*	LCJ-5-4	Culture centrifugal extractionand purification by QFF column	Glc, Man (1.7:1.0)	7 × 10^3^	Antioxidant	[86]
*Penicillium solitum*	GW-12	Ethanol precipitation, anion-exchange and size exclusion chromatography	Man	1.13 × 10^4^		[87]
*Hansfordia sinuosae*	HPA	Ethanol precipitation, anion-exchange and size exclusion chromatography	Man, Gal, Glc, (96.1, 3.3, and 0.60)	2.25 × 10^4^	Anticancer	[88]

**Table 3 marinedrugs-20-00512-t003:** Information of some EPS obtained from marine microalgae and cyanobacteria in the last decade.

Source	Name	Preparation Method	Monosaccharides Composition	Mw (Da)	Bioactivity	References
*Porphyridium sordidum*	EPS	Cold aqueous centrifugal extraction and purification by dialysis	Fuc, Rha, Ara, Gal, Glc, Xyl, GlcA (1.93:0.36:0.36:48.28: 19.01:28.2:0.76)		Antibacterial	[24]
*Porphyridium marinum*	EPS-0C, EPS-2C, EPS-5C	Culture centrifugal extraction, ultra-filtration and High-Pressure Homogenizer	Xyl, Gal, Glc, Fuc, Ara, GlcA (44–47:25–29:19–20:1:1–2:4–5)	1.4 × 10^6^ 5.5 × 10^5^ 5.5 × 10^5^	Antibacterial, Anti-biofilm, Anticancer	[95]
*Flintiella* *sanguinaria*	EPS	Culture centrifugal extraction and purification by ultra-filtration	Xyl, Gal, GlcA, Rha, Glc, Ara (47:21:14:10:6:2)	1.5 × 10^6^		[96]
*Cyanothece* sp. CCY 0110	Cyanoflan	Cold aqueous extraction and purification by dialysis	Man, Glc, uronic acid, Gal, Xyl, Rha, Fuc, Ara (20:20:18:10:9:9:8:6)	>1 × 10^6^		[75]
*Chlamydonas * *reinhardtii*	EPS	Culture centrifugal extraction	GalA, Rib, Rha, Ara, Gal, Glc, Xyl	2.25 × 10^5^	Antioxidant	[97]
*Nostoc* *carneum*	EPS	Culture centrifugal extraction	Xyl, Glc (4.3:2.1)		Antioxidant	[93]
*Nostoc* sp.	EPS	Culture centrifugal extraction and purification by DEAE column	Uronic acid, Rha, Fuc, Ara, Xyl, Man, Gal, Glc (25.0:0.2:0.8:18.6:15.3:19.1:1.3:19.7)	2.37 × 10^5^	Antitussive, Immunomodulatory activity	[98,99]
*Tetraselmis suecica*	EPS	Cold aqueous extractionand purification by dialysis	Ara, Rib, Man, GalA, Gal, Glc, GlcA (5.23:0.83:6.64:0.1:25.27:35.46:21.47)		Antioxidant, Anticancer	[94]
*Leptolyngbya* sp.	EPS	Culture centrifugal extraction	Man, Ara, Glc, Rha, uronic acid (35:24:15:2:8)		Antioxidant	[100]

## Data Availability

The data presented in this study are available on request from the corresponding author.

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
