# Peer review of "Exopolysaccharides from Marine Microbes: Source, Structure and Application"

_marinedrugs, 2022, doi:10.3390/md20080512_

Round 1
Reviewer 1 Report
The article “Exopolysaccharides from Marine Microbes: Source, Structure and Application” by Mingxing Qi et al. is a review on the progress in studies related to structure, biological activities and applications of exopolysaccharides produced by marine microorganisms.
The topic of the review is very interesting, and an important amount data is discussed.
The following sub-chapters are included in the review: (2) statistical analysis of recent studies related to EPS of marine micro-organisms; (3) Biosynthetic and structural characteristics of EPS; (5) Overview of the studies of EPS produced by marine microorganisms in the last decade; (6) Progress in the studies of on biological activities of marine EPS.
However, in my opinion, the presentation of the material lacks clarity and logic. A thorough editing is necessary prior to publication. In several places literature references are missing.
Below are a few examples.
General comment:
The scope of the review in relation to the definition of “Marine Microbes” is not clearly defined. Along with marine microorganisms, the authors also discuss data on polysaccharides from macro- and micro-algae (Ex. Lines 365-370; lines 408-412). EPS from fungi are discussed only in chapter 6.1.2 in relation to antioxidant activity.
More specific comments:
n P. 4; the description of the extraction and purification methods is very general; references are missing. Line 123-124 : the definition is wrong (homopolysaccharides).
n Page 5. Discussion of structures of EPS lacks logic and is rather confusing. There is no clear distinction between EPS with established structures and crude preparations with only preliminary information available. In my opinion, the general discussion on methodology of carbohydrate analysis (lines 148-168) is imprecise and unnecessary here.
n Table 1 contains a lot of valuable information. The title of the table is not very clear. The table itself would benefit from a better formatting (ex. Vertical lines)
n If I am not mistaken, there is an error in chapter numbering (Chapter 4 is missing).
n Chapter 5. The title should be re-formulated.
n Page 12, Chapter 6.2.1. The role of EPS in waste-water treatment. Re-structuring the paragraph would greatly benefit for a better clarity of the presentation
Author Response
Dear reviewers:
Thank you very much for your valuable comments. We agreed with your point. We carefully considered the problems you mentioned and revised the manuscript step by step by rechecking literatures, especially those in the last decade. Irrelevant content was removed and several information related were added. In response to your questions, we make the following changes:
- The revised manuscript defines Marine microorganisms as bacteria, fungi, and microalgae. Their detailed information is listed in Tables 1-3, while Figure 2 shows the structure of Marine fungi in the last decade.
- For the description of the extraction and purification methods has been carefully revised. The wrong definition of homopolysaccharides has been removed.
- According to your comments, we supplemented the information in Table 1, which added the preparation methods and the molar ratio of monosaccharides that composed EPS, and also added the information of Marine fungi, Marine microalgae and Cyanobacteria.
- The structure of the article has been readjusted, and there is no chapter sorting problem and the language was also carefully reworked and polished. Grammar errors are also carefully corrected.
Please see attached.
Sincerely,
Peipei Wang, PhD, College of Food Science and Technology, Shanghai Ocean University, China;
Guangli Yu, Professor, Key Laboratory of Marine Drugs, Ministry of Education, Shandong Provincial Key Laboratory of Glycoscience and Glycoengineering, School of Medicine and Pharmacy, Ocean University of China, Qingdao 266237, China;

Reviewer 2 Report
The review manuscript Marine Drugs 1809200 entitled “Exopolysaccharides from Marine microbes: source, structure and application” describes a relevant topic even if the review is poor and not well organised. More specifically the manuscript is unbalanced, and much space is devoted to the biological activity with respect to the chemical structure and to methodologies, of which only a picture is present. No descriptions of new methodologies can be found. No connections among the paragraphs. Many references are reviews, and not original papers, as they should be. Some of the references in Table 1 regard polymers other than EPS, such as polyglutamic acid. Moreover, the biosynthetic pathway for EPS is described in paragraph number 3 by a picture only. Some misprints are also present.
For all these reasons, in my opinion, the manuscript should be rejected.
Author Response
Dear reviewers:
Thank you very much for your valuable comments. We agreed with your point. We carefully considered the problems you mentioned and revised the manuscript step by step by rechecking literatures, especially those in the last decade. Irrelevant content was removed and several information related were added. The main modification was written in Blue in the revised manuscript. In response to your questions, we make the following changes:
- The review structure has been reoptimized. Now this review mainly foucus on the source, structure and applicaiton of marine microbial EPS in the last decade, so that in order to make it clear, the content of the newly revised manuscript changed to six parts: introduction, source of EPS, structural characterisation of EPS, biological activities of EPS, environmental remediation applications of EPS, conclusion and prospect.
- More space is devoted to chemical structure, and the review is more balanced than before. At the same time, we increased the number of original articles and reduced the number of reviews in the references.
- According to your comments that the biosynthetic pathway for EPS is described in paragraph number 3 by a picture only, we have written more comments and enriched the content of biological activity in each section.
Please see attached.
Sincerely,
Peipei Wang, PhD, College of Food Science and Technology, Shanghai Ocean University, China;
Guangli Yu, Professor, Key Laboratory of Marine Drugs, Ministry of Education, Shandong Provincial Key Laboratory of Glycoscience and Glycoengineering, School of Medicine and Pharmacy, Ocean University of China, Qingdao 266237, China;

Reviewer 3 Report
Though the topic is of great interest, this review is superfluous and useless. The novelty as well as the quantity of the information is limited. In contrast to the promise given in the title, only three structures are shown, all three were already mentioned in a review published in 2018 (Ref. [16]). Two of them were published originally much before the last decade. Otherise, only monosaccharide composition is cited. A review of the EPS from microalgae and cyanobacteria contaning many structures (doi:10.3390/md20050336) is ignored.
Besides there are some flops which should not appear in a manuscript submitted to any scientific journal.
Some references appear multiply, e.g. [1], [20], and [32] are the same, though with different style; likewise, [15], [24], and [155] are the same. I stopped checking.
I suspect plagiarism in figures, i.e.:
Figure 3 is, without indication, a marginally modified copy of Fig. 2 shown in Ref. [32].
The structures shown in Fig. 4 are copied from Fig. 4 in Ref. [16], again without appropriate citation.
Besides the weakness of the contents, there are several incorrect formulations and wrong statements, e.g.
line 81: "unique glycosidic linkage showed cytotoxicity to tumor cells" - There is nothing like this in Matsuda’s paper. Anyway, how can a glycosidic linkage show cytotoxicity?
line 108: "syntheses-dependent pathway"
line 132: "galactouronic acid"
line 133: the figure number is wrong
line 356: "Anti-infectious diseases"
line 360: "production of immune"
line 361: "infected bacteria"
line 362: "marine chitosan 20" (this wrong statement is copied from Ref. [149]. By no means, chitosan is a marine polysaccharide, even if some people say so).
line 553: "Vibriodiabolicus"
Section 6.1.4 refers much to macroalgae and plants which have nothing to do with the topic of this paper.
etc. etc.
In conclusion, this manuscript is inacceptable.
Author Response
Dear reviewers:
Thank you very much for your valuable comments. We agreed with your point. We carefully considered the problems you mentioned and revised the manuscript step by step by rechecking literatures, especially those in the last decade. Irrelevant content was removed and several information related were added. The main modification was written in Blue in the revised manuscript. Breifly, the main modifications are as follows.
- This review mainly foucus on the source, structure and applicaiton of marine microbial EPS in the last decade, so that in order to make it clear, the content of the newly revised manuscript changed to six parts: introduction, source of EPS, structural characterisation of EPS, biological activities of EPS, environmental remediation applications of EPS, conclusion and prospect.
- According to your comments that uesful information are not enought, several new information and references related are complementary. For example, three tables are added to show information of EPS obtained from marine bacteria, fungi, microalgae and cyanobacteria in the last decade, respectively. Four figures are added. Here are lists of tables and figures. In addation, the relevant content have also been supplemented in the revised manuscript.
Table 1 Information of some EPS obtained from marine bacteria in the last decade.
Table 2 Information of some EPS obtained from marine fungi in the last decade.
Table 3 Information of some EPS obtained from marine microalgae and cyanobacteria in the last decade
Fig. 1 The graph abstract of this review.
Fig. 2 The possible structure of several marine fungal EPS in the last decade.
Fig. 3 Main biological activities of EPS from marine microorganisms.
Fig. 4 Potential mechanisms of ferroptosis-related immunomodulatory of AUM-1.
- The language was also carefully reworked and polished. Grammar errors are also carefully corrected.
Please see attached.
Sincerely,
Peipei Wang, PhD, College of Food Science and Technology, Shanghai Ocean University, China;
Guangli Yu, Professor, Key Laboratory of Marine Drugs, Ministry of Education, Shandong Provincial Key Laboratory of Glycoscience and Glycoengineering, School of Medicine and Pharmacy, Ocean University of China, Qingdao 266237, China;

Round 2
Reviewer 1 Report
The manuscript was significantly improved and, in my opinion, can be published in Marine Drugs. However, there are still several errors/imprecisions in the text. Some formulations might be misleading.
I suggest to make some corrections to the text prior to publication.
In particular :
- Line 45: hydrophobicity (?) please check
- Lines 150-153. Please consider re-writing the text with more precision, here is my suggestion: Monosaccharide composition of EPS has been determined by a variety of methods including acid hydrolysis followed by appropriate derivatization and gas chromatography (GC); pre-column derivatization with high performance liquid chromatography (HPLC) and high performance anion-exchange chromatography with pulsed amperometric detection (HPAEC-PAD).
- Line 181 : Examples of Marine microbial EPS studied in the last decade
- Line 188 and line 269: Tables 1-3
- Line 217 : Theses EPS isolated from marine fungi
- Line 218 : I suggest to replace “a certain degree of structural regularity” with something like “less significant diversity in monosaccharide composition”
- Lines 229 and 234, legend to Figure 2 : Suggested structures
- Line 234, legend to Figure 2, probably : Suggested structures of several EPS, isolated from marine fungi in the last decade
Author Response
Thank you very much for your valuable comments. We agreed with your point. We have carefully revised it point by point. The new modified font is marked in red.
- hydrophobicity has now been deleted.
- Lines 150-153 has been revised according to your suggestion.
- Line 181 An EPS found in the last decade have been added and replaced the above one.
- The question of singular and plural in the article has been corrected.
- Line 218 has been revised according to your suggestion.
- Legend to Figure 2 has now been revised.
Reviewer 2 Report
The manuscript marinedrugs-1809200, describing marine exopolysaccharides and some of their applications, has been surely improved. The paper is now well organized and correctly focused.
In my opinion, it can be accepted after these revisions:
1. Purification methods have been added only in Table 1. Before the structural characterization, some sentences explaining the problems connected to the EPS purification should be added. For example, what about the elimination of proteins from the EPS sample? This should be carefully explained since the biological activity is altered by the presence of proteins.
2. SDS and DOC electrophoresis with alcian blue staining should be mentioned. This technique is very useful to detect the presence of EPS.
3. Other techniques, such as SANS and CD, with examples could be added to the physicochemical methods
Author Response
Thank you very much for your valuable advice. We have carefully revised it point by point.
- The purification methods have been added in Table 1-3. A description of the purification method is added and written in Red in the 3.1 section
- Thank you for your nice suggestions. We have added the method of SDS and DOC electrophoresis with alcian blue staining and related reference in the 3.1 section.
- Thank you for your nice suggestions. We have also added the method of SDS and DOC electrophoresis with alcian blue staining and related references in the 3.1 section.
Reviewer 3 Report
The manuscript is improved but still needs careful editing.
Table 2: Ref. [79] does not refer to Aspergillus terreus. A reference for this fungus is missing. Cite the proper publication.
Geobacillus tepidamans (ref. [131]): this fungus is from a terrestrian hot spring, not a marine source.
Figure 2, general: the names of the species are hardly readable, a larger font is necessary.
Figure 2, legend: references must be added for each of the structures, even if they are in the text and tables.
Figure 2b: the name of the fungus is misspelled (versicolao)
Figure 2e: i) the circular four carbohydrate macrocycle is not shown in this structure; ii) the sulfated unit must be written with the correct carbohydrate nomenclature, i.e. -α-d-Glcp-(1®6)-α-d-Glcp4S-(1®6)-α-d-Glcp-, see IUPAC-IUBMB rule 2-Carb-24.3. Sulfates; iii) the position of residues R1 and R2 must be shown in the backbone structure.
General:
i) Fischer stereodescriptors (d,l) must always be typed in small capitals.
ii) The name of a species following the genus must be typed always in small letters (Genus species). Double-check carefully.
Author Response
Thank you very much for your kind suggests. We have carefully revised it point by point.
Table 2: Ref. [79] does not refer to Aspergillus terreus. A reference for this fungus is missing. Cite the proper publication.
Answer:We have changed the proper reference to replace the Ref.[79].
Geobacillus tepidamans (ref. [131]): this fungus is from a terrestrian hot spring, not a marine source.
Answer: We have delated this reference[131].
Figure 2, general: the names of the species are hardly readable, a larger font is necessary.
Answer: A larger font has been used to make the names of the species clear in Fig.2.
Figure 2, legend: references must be added for each of the structures, even if they are in the text and tables.
Answer: The related references have been cited in the Fig. 2 legend.
Figure 2b: the name of the fungus is misspelled (versicolao)
Answer: The name of the fungus is corracted.
Figure 2e: i) the circular four carbohydrate macrocycle is not shown in this structure; ii) the sulfated unit must be written with the correct carbohydrate nomenclature, i.e. -α-d-Glcp-(1®6)-α-d-Glcp4S-(1®6)-α-d-Glcp-, see IUPAC-IUBMB rule 2-Carb-24.3. Sulfates; iii) the position of residues R1 and R2 must be shown in the backbone structure.
Answer: Thank you for your kind suggestions. Actually, this structure is much complex and long. To make this structure more clearer, we have tried to draw this structure in Haworth form, according to the figure in the cited reference.
General:
- i) Fischer stereodescriptors (d,l) must always be typed in small capitals.
Answer: Thank you. We checked the structure in Fig. 2 carefully to make sure they are consistent with the cited references.
- ii) The name of a species following the genus must be typed always in small letters (Genus species). Double-check carefully.
Answer: Thank you for your nice suggestions. The name of a species following the genus have been double-check carefully to make sure they are in small letters.